# A Novel Structure Harboring *bla*_CTX-M-27_ on IncF Plasmids in *Escherichia coli* Isolated from Swine in China

**DOI:** 10.3390/antibiotics10040387

**Published:** 2021-04-04

**Authors:** Yan Zhang, Yin-Huan Sun, Jiang-Yang Wang, Man-Xia Chang, Qiu-Yun Zhao, Hong-Xia Jiang

**Affiliations:** 1Guangdong Laboratory for Lingnan Modern Agriculture, Guangzhou 510642, China; yz_scau@163.com; 2Guangdong Key Laboratory for Veterinary Drug Development and Safety Evaluation, College of Veterinary Medicine, South China Agricultural University, Guangzhou 510642, China; sunyh0210@163.com (Y.-H.S.); wjy19927533599@163.com (J.-Y.W.); 13424457247@163.com (M.-X.C.); qiuyunzhao123@163.com (Q.-Y.Z.)

**Keywords:** *bla*_CTX-M-27_, Tn*2*, plasmid, *Escherichia coli*, ST10

## Abstract

The aim of this study was to elucidate the prevalence of *bla*_CTX-M-27_-producing *Escherichia coli* and transmission mechanisms of *bla*_CTX-M-27_ from swine farms in China. A total of 333 *E. coli* isolates were collected from two farms from 2013 to 2016. Thirty-two CTX-M-27-positive *E. coli* were obtained, and all were multidrug-resistant. Pulsed field gel electrophoresis (PFGE) and multilocus sequence typing (MLST) profiles indicated a wide range of strain types that carried *bla*_CTX-M-27_, and the sequence type ST10 predominated. Conjugation, replicon typing, S1-PFGE and hybridization experiments confirmed that 28 out of 32 CTX-M-27 positive isolates carried *bla*_CTX-M-27_ genes on plasmids F18:A-:B10 (16) and F24:A-:B1 (12).The *bla*_CTX-M-27_ genes for 24 isolates were transmitted by plasmids with sizes ranging from 40 to 155 kb. A comparative analysis with *bla*_CTX-M-27_-plasmids indicated that the *tra-trb* region of F24:A-:B1 plasmids was destroyed by insertion of a complex region (eight isolates) and a novel structure containing *bla*_CTX-M-27_ in the F18:A-:B10 plasmids (12 isolates). The novel structure increased the stability of the *bla*_CTX-M-27_ gene in *E. coli*. This study indicated that the predominant vehicle for *bla*_CTX-M-27_ transmission has diversified over time and that control strategies to limit *bla*_CTX-M-27_ transmission in farm animals are necessary.

## 1. Introduction

Production of extended-spectrum β-lactamases (ESBLs) is the principal mechanism of resistance to cephalosporins. CTX-Ms was the largest group of ESBLs, and they have become globally disseminated [1]. A recent study indicated that food-producing animals represent an important source of *bla*_CTX-M_-producing *E coli* isolates [2]. CTX-M-15 and CTX-M-14 are the most prevalent. Additionally, *bla*_CTX-M-27_ has been detected as increasing rapidly in prevalence [1]. The detection of CTX-M-27 in *Escherichia coli* patient isolates has been increasing, and is especially alarming because of its presence in clonal groups such as ST10, ST69 and ST131 [3,4,5,6]. In particular, *E. coli* ST131 isolates are the primary hosts responsible for CTX-M-27 transmission in *E. coli* from companion animals, the environment, farm animals and animal products [3,7,8,9,10,11,12,13]. IncF plasmids and transposons are frequently associated with transfer of *bla*_CTX-M-27_ in *E. coli* and are frequently associated with IncF plasmids and transposons [14,15]. The pressure of antibiotics could facilitate the evolution of the transmission mechanism of *bla*_CTX-M_. The prevalence of *bla*_CTX-M-27_ in *Salmonella* isolated from food animals is also increasing, and transduction of *bla*_CTX-M-27_ in *Salmonella* isolated from pork has been recently demonstrated to be mediated by a P1-like bacteriophage that had integrated a Tn*1721*-like structure [16]. A prospective study utilizing salmonella and *E. coli* isolates from 2003 to 2014 indicated that *bla*_CTX-M-27_ could be mobilized via a Tn*1721*-like structure between *E. coli* plasmids and P1-like bacteriophage in salmonella [17]. 

Intestinal carriage of CTX-M-producing bacteria (especially pathogenic bacteria) in food-producing animals and retail meat or dairy products contaminated by CTX-M β-lactamases might contribute to increased occurrence of infections with ESBL-producing bacteria in humans [18]. The prevalent vehicle for *bla*_CTX-M_ transmission has diversified over time. This would lead to the further spread of the *bla*_CTX-M-27_ gene and further highlights the importance of controlling the spread of *bla*_CTX-M-27._ In the current work we investigated the prevalence and transmission mechanisms of CTX-M-27 *E. coli* from swine in Guangdong province, China from 2013 to 2016. We identified a novel structure containing *bla*_CTX-M-27_ that contributed to the stability of *bla*_CTX-M-27_ in *E. coli*.

## 2. Results and Discussion

### 2.1. Phenotypes of bla_CTX-M-27_-Carrying Isolates

We surveyed two swine farms in Guangdong province and confirmed the presence of *bla*_CTX-M-27_ in 32 *E. coli* isolates including 62.5% (20) from farm 1 and 37.5% (12) from farm 2 (Figure 1). CTX-M-27 has been only sporadically detected in *E. coli* isolates from food-producing animals, and its prevalence has remained constant at <8% over the past decade [17,19]. The prevalence of *bla*_CTX-M-27_ in our study (9.61%) was higher than in previous studies. Importantly, all 32 CTX-M-27 isolates were multidrug-resistant (MDR) strains (Appendix A). The prevalence of ciprofloxacin-resistance in these isolates was 93.75% compared with 77.78% from isolates in 2003 to 2009 in China on swine farms [17], and the prevalence of ciprofloxacin resistance in *bla*_CTX-M-14_ or *bla*_CTX-M-15_-producing *E. coli* almost above 80% [20,21]. Additionally, we detected QRDR (Quinolone Resistance Determining Regions) mutations in *gyrA* (S83L and D87N) and *parC* (S80I) in high-level quinolone resistant isolates (28), and 14 of these coharbored *oqxAB* (Figure 1). This was consistent with a previous study where blood isolates of *bla*_CTX-M-27_-producing ST131 *E. coli* possessed QRDR mutations that enabled high level resistance to quinolones [22]. Similarly, *bla*_CTX-M-14_, *bla*_CTX-M-55_, *bla*_CTX-M-15_ and *bla*_CTX-M-9_-producing *E. coli* possessed QRDR mutations and carried PMQR genes that enabled high level resistance to quinolones [23]. In contrast, our isolates possessed a lower MIC_CIP_ range of 0.25 to 1 μg/mL but did not contain QRDR mutations although either *qnrS* or *oqxAB* were present. This was consistent with a previous study reporting that *bla*_CTX-M-55_-producing salmonella with an MIC_CIP_ in the 0.5 to 4 μg/mL range did not contain QRDR mutations but all contained *qnrS* [24]. In addition, 9/32 (28.1%) of our isolates coharbored *mcr-1*, and these exhibited resistance to colistin at MIC = 8 μg/mL (Figure 1).

The coexistence or cotransfer of other resistance genes in CTX-M-producing *E. coli* strains increases their probability of survival in the presence of other antibiotics, including the cephalosporins [24,25,26]. Infections caused by MDR bacteria are becoming common and represents a serious public health concern [27].

### 2.2. Molecular Typing Analysis

In our group of 32 CTX-M-27-producing *E. coli* isolates, we identified seven sequence types (ST). The ST131 was previously found to be associated with *bla*_CTX-M-27_ in Japan, North America and Europe [28,29,30]. However, ST131 was not present in any of our isolates. We found that ST10 was most prevalent (50%, 16/32) followed by ST224 (28.1%, 9/32), ST101 (9.4%, 3/32). ST10 and ST224 were detected on both farms. The isolates containing ST46, ST162, ST93 and ST58 were represented by one isolate each (Figure 1). Recent reports documented that ST10 and ST224 *E. coli* are primarily carriers of CTX-M-15, CTX-M-14, CTX-M-1 and CTX-M-8 [31,32,33,34,35,36,37]. Our recent study identified ST8900 as the most prevalent ST (Sequence Typing) in CTX-M-27-positive *E. coli* isolates obtained during 2003 to 2009 [17]. These results indicate that the dominant ST of *bla*_CTX-M-27_-producing *E. coli* has changed. ST10 *E. coli* contributes to the distribution of the *bla*_CTX-M-27_ gene on swine farms in China. Previous studies showed that although the epidemiology and population dynamics of Global Extraintestinal Pathogenic *E. coli* (ExPEC) are complex, ST10 was the major ST the ExPEC (Extraintestinal Pathogenic *Escherichia coli*) group, and has remained at a constant level. ST10 has now been linked to multidrug resistance gene carriage or ESBL production [38]. Increased numbers of antibiotic resistance genes have been found and are carried by pathogenic bacteria and this is a human health concern. The 32 CTX-M-27 isolates we found could also be divided into 14 different PFGE clusters. These data indicated that the transmission of *bla*_CTX-M-27_ among *E. coli* isolates occurred via clonal expansion and horizontal transmission. The results of both clonal spread of resistant strains and horizontal transmission of the resistance plasmids contributed to the dissemination of *bla*_CTX-M-27_-positive salmonella or *E. coli* isolates were also obtained in previous studies [16,17,39].

### 2.3. Location and Genetic Context of the bla_CTX-M-27_ Gene

The genomic locations of the *bla*_CTX-M-27_ for our 32 isolates included F plasmids of 40–155 kb for 28 isolates and four were present on nontypeable incompatibility group plasmids. Most of these endogenous plasmids (24. 75%) could be successfully transferred to a laboratory strain *E. coli* C600 by conjugation. This was consistent with previous studies reporting that IncF plasmids are frequently associated with *bla*_CTX-M_ genes [17]. Plasmid multilocus sequence typing (pMLST) of the IncF plasmids indicated only two plasmid types: F18:A-:B10 (16) and F24:A-:B1 (12) (Figure 1).

The plasmid p1-8 (F18:A-:B10) was completely sequenced (155,669 bp), possessed a G + C content of 51.56% and possessed a full complement of DNA transfer genes in the F18:A-:B10 backbone (Figure 2B). These transfer regions were highly similar to other F plasmids and differed by the number of CAACAGCCG tandem repeats in the *traD* gene. Our isolates all possessed only one of these repeats and were highly similar to that of a *bla*_CTX-M-__55_-harboring F18:A-:B1 plasmid pCREC-544_1 (NZ_CP024827.1) from a human clinical isolate from Korea, and a *bla*_NDM-5_-harboring F18:A-:B58 plasmid pMTY18780-2 (NZ_AP023199.1) human isolate from Japan. IncF plasmids have been implicated as main players for transmission of *bla*_CTX-M-27_ [40]. So, we estimate that F18:A-:Bx plasmids are the primary vectors for animal to human transmission of most antibiotic resistance genes, and this has significant health implications for both humans and animals.

The second plasmid pM8-1 (F24:A-:B1) we identified possessed DNA transfer genes that had been altered by insertion of a 17,034 bp region that included Tn*As3*, IS*5075*, IS*26*, *intI1*, *mph* and IS*6100*, as well as two IS*26* copies in the same orientation. This insertion destroyed *traI* and *traN*, and the entire *trbH*/*F*/*G*/*B*/*A*/*E*, *traD*/*T*/*S*/*G*/*H*/*Q*/*F* region was lost (Figure 2). A similar disruption was previously documented for pHNAH9 that lacked most parts of the *tra-trb* region due to multiple recombination events, and this most likely accounted for its conjugation failure. Our F24:A-:B1 plasmids were highly similar to pHNAH9 [41]. These results indicate that *E. coli* ST224 possessing F24:A-:B1 was disseminated prior to its acquisition of *bla*_CTX-M-27_, similar to the *bla*_CTX-M_-carrying F1:A2:B20 plasmid in *E. coli* ST131-H30R1 [42].

The region (Figure 3a) containing *bla*_CTX-M-27_ (17354 bp) was present in both our plasmid types, F18:A-:B10 and F24:A-:B1. This region was identical to the variable region from pMRY15_117 (accession no: NZ_AP017618.1, Figure 3a). The insertion of IS*Ecp1*-*bla*_CTX-M-27_-IS*903B* in the *tnpA* transposition gene of Tn*2* might be expected to prevent transposition. However, IR_TEM_ had been lost, although the IR_tnp_ and the resolvase (*res*) sites were intact. Therefore, it is possible that the TnpA function was provided by another Tn*3*-like transposon and that this mediated movement of this interrupted Tn*2* region [43]. Thus, Tn*2*-mediated events contributed to the spread of *bla*_CTX-M-27_ in the F24:A-:B1 plasmids.

We also identified another genetic context contained within a 22,178 bp insertion (Figure 3b). This was a novel structure associated with *bla*_CTX-M-27_. The region responsible for *bla*_CTX-M-27_ transfer possessed a common background region ΔIS*Ecp1*-*bla*_CTX-M-27_-IS*903B* within the variable region. The region downstream of IS*903B* was the 312 bp Δ*tnpA* of Tn*2* produced by the insertion of IS*15DI* and Tn*As3* in pSCU-103 (CP054458.1) (Figure 3d) and differed from p53_A (Figure 3e) (684 bp Δ*tnpA/res/tnpR* of Tn*2*). In p4_4.1 (CP023827.1) (Figure 3c), the insertion of IS*15* resulted in Δ*tnpR/res/tnpA*Δ*-iroN*Δ inversion. The *iroN* and Tn*2* sequences were lost, and R1, R2, R3 and *aadA5*-*qacE*Δ*1*-*sul1* were all inverted in p1-8, while *intI1* and *dfrA17* from In*54* were lost (Figure 3b). Taken together, these results indicate that Tn*2* was incomplete and incapable of transfer of *bla*_CTX-M-27_ in these plasmids. Therefore, F18:A-:B10 was one of the primary contributors to the spread of *bla*_CTX-M-27_ in *E. coli* for our swine isolates and resulted in greater stability of *bla*_CTX-M-27_ in clinical isolates. This indicates that the *bla*_CTX-M-27_ gene was stable in vehicles and its natural host. The plasmid can mediate its transmission in a wider range hosts, which poses a serious threat to public health. The IS*Ecp1*-like family has been shown to act as a promoter region for high-level expression of different CTX-M enzymes in a previous study [44]. The conserved region responsible for transmission of *bla*_CTX-M-27_ was IS*ECP1*-*bla*_CTX-M-27_-IS*903B*. Since more complex structures that mediate transmission of *bla*_CTX-M-27_ have been recently described, and a diversity of media promotes the transmission of *bla*_CTX-M-27,_ it may cause a serious threat to public health.

The F18:A-:B10 plasmids may make it easier for *E. coli* ST10 to acquire *bla*_CTX-M-27__,_ and similar results have been reported for the MDR IncF group F1:A2:B20 plasmid in *E. coli* ST131-H30R1 [42]. Our recent work revealed that *bla*_CTX-M-27_-plasmids isolated from 2003 to 2009 contained a genetic context with the Tn*1721*-like structure ΔIS*Ecp1B*-*bla*_CTX-M-27_-IS*903D*-*iroN*-Δ*map*-Tn*1721* [17]. In contrast, the present study indicated that the horizontal transmission of *bla*_CTX-M-27_ was related to Tn*2*, so that the prevalent vehicle for *bla*_CTX-M-27_ transmission has diversified over time.

## 3. Materials and Methods

A total of 333 *E. coli* isolates were isolated from swine swabs from Guangdong province, China and were maintained in our laboratory. These were obtained between June 2013 and September 2016 from farm 1 (95/236, 40.25%) and farm 2 (141/236, 59.75%). The presence of the *bla*_CTX-M-27_ gene was confirmed by PCR (Polymerase Chain Reaction) and DNA sequencing [45]. The minimum inhibitory concentration (MIC) of *bla*_CTX-M-27_-positive isolates was determined for the following: ampicillin (AMP), ceftiofur (CTF), cefotaxime (CTX), ceftazidime (CTZ), meropenem (MRO), florfenicol (FLF), chloramphenicol (CHL), gentamycin (GEN), streptomycin (STR), amikacin (AMI), apramycin (APR), ciprofloxacin (CIP), nalidixic acid (NAL), dafloxacin (DF), olaquindox (OQX), tetracycline (TET), fosfomycin (FOS), enrofloxacin (ENR) and kanamycin (KAN), determined by the agar dilution method, and colistin (CL) was determined by the microdilution broth method. *Escherichia coli* ATCC 25,922 was used as a quality control strain. The results were interpreted following Clinical and Laboratory Standards Institution (CLSI) guidelines (2015, M100-S25) and veterinary CLSI (VET01-A4/VET01-S2). 

All *bla*_CTX-M-27_ positive isolates were screened for the presence of the β-lactamase genes (*bla*_TEM_, *bla*_SHV_, *bla*_CTX-M-1G_, *bla*_CTX-M-9G_), carbapenem resistance genes (*bla*_IMP_, *bla*_VIM_, *bla*_NDM_, *bla*_OXA_) and PMQR genes (*qnrA*, *qnrB*, *qnrC*, *qnrD*, *qnrS*, *oqxAB*, *aac(6**‘)-Ib-cr*, *qepA*) using previously described primers and protocols [18]. The DNA sequences and deduced amino acid sequences were compared with sequences available at GenBank using BLAST (Basic Local Alignment Search Tool) https://blast.ncbi.nlm.nih.gov/Blast.cgi (*bla*_CTX-M-27_-bearing plasmids from the NCBI nonredundant database were retrieved as of 15 February 2021). Mutations in the QRDR of the target genes *gyrA*, *gyrB*, *parC*, *parE* were confirmed by PCR and sequencing, and compared with the *E. coli* K12 genome as a reference. 

MLST was performed for *bla*_CTX-M-27_-producing isolates as previously reported [46]. The PCR and DNA sequence analysis of seven housekeeping genes (*adk*, *fumC*, *gyrB*, *icd*, *mdh*, *purA* and *recA*) was performed, the allelic profiles were determined, and the sequence type (ST) was assigned in accordance with http://mlst.warwick.ac.uk/mlst/dbs/, (accessed on 15 February 2021). Pulsed field gel electrophoresis (PFGE) patterns with the *XbaI* enzyme were analyzed in experiments following the method of Jiang et al. for genetic relatedness of all *bla*_CTX-M-27_-harboring isolates [18]. 

*E. coli* C600 strain (streptomycin-resistant) was used as donor strain and clinical *bla*_CTX-M-27_-producing strains were used as recipient strains to test the transferability of the *bla*_CTX-M-27_ gene. Transconjugants were selected on MacConkey agar supplemented with 1 mg/L cefotaxime and 2000 mg/L streptomycin [18]. PCR-based replicon typing (PBRT) was performed on all transconjugants with *bla*_CTX-M-27_-carrying plasmids using primers as previously described [47]. To determine the location of *bla*_CTX__–M__–27_, all the transconjugants were subjected to pulsed-field gel electrophoresis (PFGE) with S1 nuclease (Takara) and Southern transfer and probing with a digoxigenin-labelled probes specific for the *bla*_CTX-M-9G_ gene as described previously [16]. 

Genomic DNA samples from M8-1 and transconjugant C-1-8 were subjected to 250-bp paired-end whole-genome sequencing using the Illumina HiSeq system (Illumina, San Diego, CA, USA), and the paired-end Illumina reads were assembled by SOAPdenovo version 2.04 [48]. To obtain the complete sequence of p1-8, PCR and Sanger sequencing was applied to close all the suspected gaps. Sequence comparisons of M8-1 and pMRY15_117 (NZ_AP017618.1) were performed using BLAST (http://blast.ncbi.nlm.nih.gov (accessed on 15 February 2021)) and Mauve [49], and similar contigs were extracted from the assemblies.

Specific primers used for further analysis of genomic context sequence on the other plasmids carrying *bla*_CTX-M-27_ gene were designed using Primer Premier v5.0 (IS1-8-fw: TCCGACACGATAGAAGGAAT/IS1-8-rev: TTGCCATCACGACTGTGC; ISM8-1-fw: CCGTCAGAAGTAAGTTGGC/ISM8-1-rev: GTGGACTGTGGGTGATAAGA). Target sequences were identified using PCR (30 cycles of denaturation at 94 °C for 30 s, annealing at 55 °C (p1-8)/52.9 °C (pM8-1) for 30 s and extension at 72 °C for 4 min 30 s. This was followed by an additional 10 min extension at 72 °C) and amplicon identities were confirmed by Sanger sequencing [50]. Gene prediction and annotation of contigs were performed using the RAST (Rapid Annotation using Subsystem Technology) server (http://rast.nmpdr.org/ (accessed on 15 February 2021)) [51] and BLAST was used for a manual review of results. The ISfinder program (https://www-is.biotoul.fr/ (accessed on 15 February 2021)) [52] and ResFinder (https://cge.cbs.dtu.dk//services/ResFinder/ (accessed on 15 February 2021)) were used to identify mobile elements and resistance genes. The replicon types of plasmids were analyzed using the plasmid MLST database (http://pubmlst.org/plasmid/ (accessed on 15 February 2021)).

## 4. Conclusions

In this study we identified ST10 *E. coli* as one of the dominant vehicles for *bla*_CTX-M-27_ transmission in China. We report a novel structure containing *bla*_CTX-M-27_ that contributed to the persistence of this gene in *E. coli*. This study identified multidrug-resistant IncF group F18:A-:B10 plasmids that contributed to the spread of *bla*_CTX-M-27_ in *E. coli*. *E. coli* ST224 possessing F24:A-:B1 were likely transmitted before acquiring *bla*_CTX-M-27_, and Tn*2*-related events may have contributed to the spread of *bla*_CTX-M-27_ in F24:A-:B1 plasmids. Additionally, the prevalent vehicle for *bla*_CTX-M-27_ transmission has diversified over time so that that transmission of the *bla*_CTX-M-27_ gene in farm animals should be closely monitored.

## Figures and Tables

**Figure 1 antibiotics-10-00387-f001:**
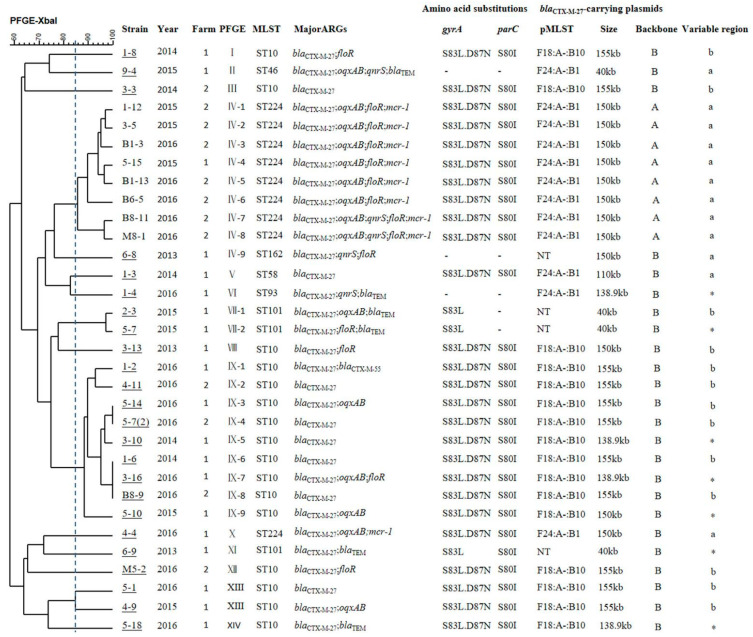
Characteristics of 32 CTX-M-27-carrying *E. coli* isolated from 2013 to 2016 in Guangdong, China. The *bla*_CTX-M-27_ genes were all located on plasmids. Isolate names that are underlined indicate that *bla*_CTX-M-27_ plasmids could be transferred to strain C600 by conjugation. *: Only the ΔIS*ECP1*-*bla*_CTX-M-27_-IS*903B* region was sequenced. Pulsed field gel electrophoresis (PFGE) patterns with a cutoff at 85% similarity (the dashed line) are considered to be the same PFGE cluster and are indicated as groups I–XIV respectively.

**Figure 2 antibiotics-10-00387-f002:**
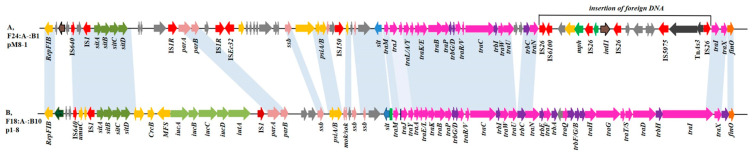
Linear comparisons of backbones for the two plasmids identified in this study. Regions of >99% identity are shaded in blue. The fluorescent pink arrows indicate *tra* area and the purple arrow indicate the *trb* area.

**Figure 3 antibiotics-10-00387-f003:**
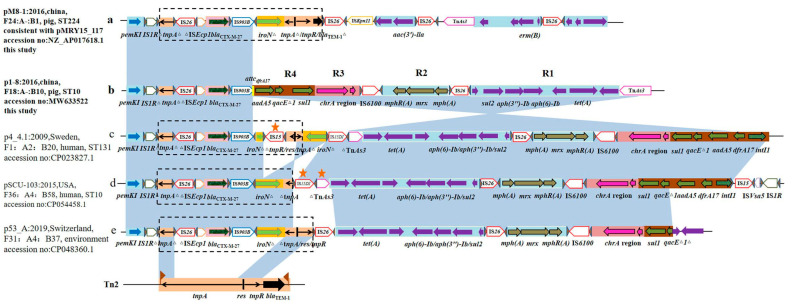
Genomic context of *bla*_CTX-M-27_ in pM8-1 and p1-8 and gene structure comparisons of with other plasmids as indicated Regions of >99% identity are shaded in blue. R1, *orf1*-*sul2*-*aph(3**′’)-Ib*-*aph(6)-Ib*-*orf2*-*tet(A)*-*orf3*; R2, *mphR(A)*-*mrx*-*mph(A)*; R3, *chrA* region; R4, *aadA5*-*qacE*Δ*1*-*sul1*. The five-pointed star indicates the insertion element. **a**: genomic context of *bla*_CTX-M-27_ in pM8-1 (F24:A-:B1, this study). **b**: genomic context of *bla*_CTX-M-27_ in p1-8 (F18:A-:B10, this study). **c**: genomic context of *bla*_CTX-M-27_ in p4_4.1 (F1:A2:B20, accession no: CP023827.1). **d**: genomic context of *bla*_CTX-M-27_ in pSCU-103 (F36:A4:B58, accession no: CP054458.1). **e**: genomic context of *bla*_CTX-M-27_ in p53_A (F31:A4:B37, accession no: CP048360.1).

## Data Availability

The complete nucleotide sequence of plasmids p1-8 have been deposited to the GenBank database and assigned accession number MW633522.

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
