# Peer review of "A Novel Structure Harboring blaCTX-M-27 on IncF Plasmids in Escherichia coli Isolated from Swine in China"

_antibiotics, 2021, doi:10.3390/antibiotics10040387_

Round 1

Reviewer 1 Report

The authors describe a novel structure containing blaCTX-M-27 which increases its stability in E. coli.  The findings suggest that the transmission was Tn2 related.  The data presented seems to support author’s conclusions. 

I would like to see what impact this new mechanism of transmission of blaCTX-M-27 would have on the animal food industry or in consequence transmission to humans. 

Also, manuscript should be revised by a person fluent in English and care should be taken with species names (see minor comments) as I found the error multiple times. 

Minor comments:

1) Lines 43 and 44: E. coli should be in italics

2) Line 52: E. coli should be in italics

3) Line 84: E. coli should be in italics

4) Line 103 and 105: E. coli should be in italics

Author Response

Dear reviewer:

On behalf of my co-authors, we are very grateful to you for giving us an opportunity to revise our manuscript. we appreciate you very much for your positive and constructive comments and suggestions on our manuscript entitled “A Novel Structure harboring blaCTX-M-27 on IncF plasmids in Escherichia coli isolated from swine in China” (Manuscript ID: antibiotics-1157308).

I am very grateful to your comments for the manuscript. According with your advice, we amended the relevant part in manuscript. The following are the responses and revisions I have made in response to the reviewers' questions and suggestions on an item-by-item basis. Thanks again to the hard work of the editor and reviewer!

Comment No. 1: I would like to see what impact this new mechanism of transmission of blaCTX-M-27 would have on the animal food industry or in consequence transmission to humans. 

Response: Thank you for your valuable comment. The novel structure enables a more stable presence of blaCTX-M-27 on the plasmid, which further demonstrates that antibiotic-resistant genes can be stable in food-producing animals, animal production or even transmitted to humans through the food chain, which poses a serious threat to public health. We supplemented this section to the manuscript and highlighted on lines 128-132, lines 166-174.

Comment No. 2: Also, manuscript should be revised by a person fluent in English and care should be taken with species names (see minor comments) as I found the error multiple times.

1) Lines 43 and 44: E. coli should be in italics

2) Line 52: E. coli should be in italics

3) Line 84: E. coli should be in italics

4) Line 103 and 105: E. coli should be in italics

Response: Thanks to Reviewer for reminder. We changed E.coli in italics on Lines 58, 69, 117, 140, 142.

Considering the Reviewer’s suggestion, we would take great effort to modify the manuscript to make it more professional. The portion of language modification are marked in yellow in the revised manuscript, and I hope it can meet with requirement.

According to the reviewer’s comments, we have revised the manuscript extensively. If there are any other modifications we could make, we would like very much to modify them and we really appreciate your help. “Antibiotics” is a journal of great popularity and prestige. We hope that our manuscript could be considered for publication in your journal. Thank you very much for your help.

Reviewer 2 Report

Majors revisions:

Although, in the style of the paper, I understand the reason for writing such a short introduction I think some other detail can be added.

In the same way, the discussion is limited to only few phrases mixed into the results/discussion section. I suggest to split the results separately from the discussion.

The sequence methods are poorly described. All the parameters used for bioinformatics analyses are missed. 

Minor revisions:

The mobile elements have been searched but no evidence appears in the figures. Are they not present at all in the investigated regions?

Supplementary table S1 is difficult to read. A table with checkmarks should be easy to interpret. 

Supplementary table 2 is not needed. The couple of used primers can be inserted in the "materials and methods" section.

Line 100-101 Revise grammar

Lines 109-110 and 120-121 - the genomic context needs to be removed from the text. The text appears to have a messy style here.

In figure 3 capture please refer to "genomic context" or at least to synthetic regions instead of "genetic environment".

Author Response

Dear reviewer:

On behalf of my co-authors, we are very grateful to you for giving us an opportunity to revise our manuscript. we appreciate you very much for your positive and constructive comments and suggestions on our manuscript entitled “A Novel Structure harboring blaCTX-M-27 on IncF plasmids in Escherichia coli isolated from swine in China” (Manuscript ID: antibiotics-1157308).

I am very grateful to your comments for the manuscript. According with your advice, we amended the relevant part in manuscript. The following are the responses and revisions I have made in response to the reviewers' questions and suggestions on an item-by-item basis. Thanks again to the hard work of the editor and reviewer!

Comment No. 1: Although, in the style of the paper, I understand the reason for writing such a short introduction I think some other detail can be added.

Response: As Reviewer suggested that it is indeed better to give other detail in introduction. There are some points which I have added to the section of Introduction (lines 26-31, lines 37-38, lines 45-50). 

Comment No. 2: In the same way, the discussion is limited to only few phrases mixed into the results/discussion section. I suggest to split the results separately from the discussion.

Response: We feel great thanks for your professional review work on our article. According to your nice suggestions, we have made extensive corrections to results and discussion of our previous draft. We have enriched the discussion and analyzed the results obtained in the manuscript more comprehensively and clearly (lines 61-62; lines 64-66; lines 70-72; lines 74-76;lines 101-106; lines 109-112; lines 128-132; lines 135-140; lines 166-174) .

However, we think that mixing the results with the discussion could explain the conclusion more intuitively. Additionly, we consider that it can facilitate readers to better compare the differences in the results of this manuscript with the previous studies and understand clearly (such as the prevalent vehicle for blaCTX-M-27 transmission has diversified over time). So we are more inclined to analyze the results and discussion together in this brief report. I look forward to reviewer's understanding with regard to the issue.

Comment No. 3: The sequence methods are poorly described. All the parameters used for bioinformatics analyses are missed.

Response: We are very sorry for our negligence of poorly describing the sequence methods and the parameters used for bioinformatics analyses are missed.We added more details about the sequence methods and the parameters used for bioinformatics analyses and they are marked in yellow in the revised manuscript. (lines 242-249, lines 250-262) .

Comment No. 4: The mobile elements have been searched but no evidence appears in the figures. Are they not present at all in the investigated regions?

Response: We feel sorry that we did not provide clear explanation about the mobile elements in the figures. The mobile elements (IS26, IS6100, IS5075, TnAs3) of foreign DNA were marked with red or black arrows in Figure 2; The incomplete Tn2 (tnpA, tnpR, blaTEM) was marked with the black arrows in Figure 3. Owing to show the structure of Tn2 more clearly, we have added dashed boxes in the figure 3. The remaining mobile elements (IS26, IS6100, IS15DI, TnAs3, IS1R, ISEcp1, IS903) were marked with red/purple pentagons.

Comment No. 5: Supplementary table S1 is difficult to read. A table with checkmarks should be easy to interpret.

Response: We are sorry for our negligence of messyly writing the Supplementary table S1. We abbreviated the antibiotic names to make the table clearer and added more detailed notes (methods of the antibiotics susceptibility test and reference criteria for results of antibiotic susceptibility tests; the full names corresponding to the abbreviation of antibiotics).

Comment No. 6: Supplementary table 2 is not needed. The couple of used primers can be inserted in the "materials and methods" section.

Response: Thank you for your valuable commen. We deleted Supplementary Table 2 and added sequences of primers and PCR conditions to materials and methods (lines 251-256).

Comment No. 7: Line 100-101 Revise grammar

Response: Thanks for your help. We feel really sorry for our carelessness.

We revised the grammar and the changes were highlighted within the document by using yellow colored  text.  (lines 127-130)

Comment No. 8: Lines 109-110 and 120-121 - the genomic context needs to be removed from the text. The text appears to have a messy style here.

Response:Thank you for your nice comments on our article. According to your suggestions, we have replaced the structural description of genomic context with Figure 3a or Figure 3b here.(lines 146-147, lines 154-155)

Comment No. 9: In figure 3 capture please refer to "genomic context" or at least to synthetic regions instead of "genetic environment".

Response:Thanks for your suggestions.we are sorry that we have not expressed 

it clearly.We have corrected the genetic environment into genomic context. (line 200)

According to the reviewer’s comments, we have revised the manuscript extensively. If there are any other modifications we could make, we would like very much to modify them and we really appreciate your help. “Antibiotics” is a journal of great popularity and prestige. We hope that our manuscript could be considered for publication in your journal. Thank you very much for your help.

Round 2

Reviewer 2 Report

I noticed all the requested changes/corrections have been made and I acknowledge the paper is now ready to be accepted for publication on Antibiotics.